# MAC: A unified framework boosting low-resource automatic speech recognition

## Abstract

We propose a unified framework for low-resource automatic speech recognition tasks named meta-audio concatenation (MAC). It is easy to implement and can be carried out in extremely low-resource environments. Mathematically, we give a clear description of the MAC framework from the perspective of Bayesian sampling. We propose a broad notion of meta-audio sets for the concatenative synthesis text-to-speech system to meet the modeling demands of different languages and different scenarios when using the system. By the proper meta-audio set, one can integrate language pronunciation rules in a convenient way. Besides, it can also help reduce the difficulty of force alignment, improve the diversity of synthesized audios, and solve the "out of vocabulary" (OOV) issue in synthesis. Our experiments have demonstrated the great effectiveness of MAC on low-resource ASR tasks. On Cantonese, Taiwanese, and Japanese ASR tasks, the MAC method can reduce the character error rate (CER) by more than **15%** and achieve comparable performance to the fine-tuned wav2vec2 model. Among them, it is worth mentioning that we achieve a **10.9%** CER on the Common Voice Cantonese ASR task, leading to about **30%** relative improvement compared to the wav2vec2 (with fine-tuning), which is a new SOTA.

## 1 Introduction

Automatic speech recognition (ASR) is a traditional task with extensive use in applications. Before the popularity of deep learning methods, HMM-GMM (R., 1989) models, possessing an elegant mathematical form, are widely adopted for ASR tasks with satisfied performance in relatively simple speech recognition tasks. The most famous HMM-GMM speech recognition toolkit is Kaldi (Povey et al., 2011). As the rise of deep neural networks (NN), a large number of end-to-end (NN-based) speech recognition models has emerged, e.g., Speech-transformer (Dong et al., 2018), Conformer (Gulati et al., 2020), LAS (Chan et al., 2015), and corresponding toolkits such as Espnet (Watanabe et al., 2018) and Wenet (Yao et al., 2021). Furthermore, there are also many pretrained large models, such as vq-wav2vec (Baevski et al., 2019), wav2vec (Schneider et al., 2019) and wav2vec2 (Baevski et al., 2020; Conneau et al., 2020).

However, whether for the HMM-GMM (R., 1989; Rodríguez et al., 1997) model or advanced end-to-end models such as Speech-transformer (Dong et al., 2018), learning a practical model for speech recognition often requires a large amount of data (hundreds of hours even tens of thousands of hours). However, in many scenarios such as dialects and minority languages, it is often difficult and expensive to get sufficient audio data for training. A straightforward solution is to use the text-to-speech (TTS) methods for data augmentation. There has been a lot of work on TTS data augmentation methods for ASR tasks, such as Laptev et al. (2020); Rossenbach et al. (2020); Sun et al. (2020). In this work, we propose a new framework called MAC (meta-audio concatenation) that can enhance low-resource ASR tasks. The MAC framework is built upon a clear mathematical demonstration from a Bayesian sampling perspective. It uses a novel concatenative synthesis TTS system that incorporates the concept of meta-audios, which are fundamental modeling units of language pronunciations. By leveraging this innovative approach, MAC can boost the performance of low-resource ASR tasks. It is worth mentioning that compared with former TTS methods for ASR tasks, our MAC framework has the following advantages:

- MAC leverages a novel concatenative synthesis text-to-speech system to boost the ASR tasks, which can integrate language pronunciation rules as the prior knowledge. Furthermore, the process of generating audios shows strong interpretability, controllability and is easy to be adjusted;

- The proposed meta-audio set is a broad notion for the concatenative synthesis text-to-speech system, and the dimensions of meta-audio sets can be flexibly determined according to prior knowledge, model complexity budgets, audio data size, etc. Using a proper meta-audio set, MAC owns benefits such as reducing difficulty in force alignment, integrating different prior knowledge and flexible applications to different scenarios with increasing diversity in synthesized audios, etc.

- There is no need to train additional TTS neural networks, and the generation process is just simple splicing, which is easy to implement and saves computation resources;

- Most importantly, the MAC framework can be carried out in extremely low-resource environments (e.g., training data less than 10 hours) without the help of additional labeled data. MAC also has a promising potential to model any low-resource languages as long as there is prior knowledge of the target language (e.g., pronunciation rules).

We perform extensive experiments to demonstrate the great effectiveness of MAC on low-resource ASR tasks. For Cantonese, Taiwanese and Japanese ASR tasks, MAC can reduce the CER by more than **15%** (Cantonese: from 32.5 to 12.7; Taiwanese: from 51.3 to 22.0; Japanese: from 45.3 to 25.0), see Table 3 for more details. Furthermore, MAC outperforms the fine-tuned wav2vec2 on the Cantonese Common Voice dataset and obtains quite competitive results on the Taiwanese and Japanese Common Voice datasets. Besides, with attention rescore decode mode, we achieves a **10.9%** CER on the Common Voice Cantonese ASR task, resulting in a significant relative improvement of about **30%** compared to fine-tuning the wav2vec2 model. Notably, these remarkable improvements in accuracy are achieved even without careful tuning of hyper-parameters.

Commonly, semi-supervised learning and transfer learning are widely utilized in low-resource speech recognition scenarios. However, both of the approaches will possibly face certain difficulties in extremely low-resource scenarios. We will briefly discuss these points below.

## 1.1 Semi-supervised learning

For semi-supervised learning, pseudo-label algorithms and their variants (e.g., iterative pseudo label algorithm (Xu et al., 2020)) are widely adopted, whose basic ideas are using the pseudo labels to help train the model. Table 1 reported in Higuchi et al. (2022) demonstrated the effectiveness of iterative pseudo-label semi-supervised algorithms. The experiments are conducted on the LibriSpeech dataset (Panayotov et al., 2015) and TEDLIUM3 dataset (Hernandez et al., 2018). Here, LL-10h, LS-100h, LS-360h and LS-860h represent different splits of the LibriSpeech dataset, and the results are evaluated on dev-other split and test-other split of the dataset.

| Resource | Dev | Test |
|---|---|---|
| LS-100h | 22.5 | 23.3 |
| LS-100h/LS-360h | 15.9 | 15.8 |
| LS-100h/LS-860h | 13.9 | 14.2 |
| LS-100h/TEDLIUM3 | 18.9 | 18.5 |
| LL-10h | 50.6 | 51.3 |
| LL-10h/LS-360h | 35.4 | 36.1 |
| LL-10h/LS-860h | 33.5 | 34.4 |

Table 1: Results of the iterative pseudo-label-based semi-supervised learning for ASR tasks (Higuchi et al., 2022). The experiments are performed on the LibriSpeech and TEDLIUM3 dataset with a CER evaluation criteria.

Although the performance can be significantly improved through the use of semi-supervised learning, we have observed that the quality of the initial models, trained on LL-10h and LS-100h data, has a remarkable impact on the final performance. As evident from the data provided in the table, a higher quality initial model, characterized by lower error rates associated with LS-100h training resource compared to LL-10h resource, leads to improved performance in the semi-supervised results, even when the same amount of unlabeled data is used. Furthermore, it is worth noting that the benefits of adding additional unlabeled data diminish as the amounbt of unlabeled data increasing. For example, the improvement obtained by adding 860 hours of unlabeled data is only approximately 2% compared to adding 360 hours of unlabeled data. This further emphasizes the significance of the quality of the initial model in influencing the final results. This observation suggests that the performance of the initial model sets an upper limit for the effectiveness of semi-supervised learning. When starting with extremely low resources, such as only 10 hours of data, a poor initial model is obtained, making it challenging for semi-supervised learning to achieve satisfactory results. There are also some theoretical results to explain these phenomena (see e.g., Wei et al. (2020) and Min & Tai (2022)). Actually, in real low-resource scenarios, it can be rather difficult to obtain a suitable initial model (or pseudo-label generator) due to the limited amount of labeled data. These difficulties seriously affect the performance of pseudo-label-based semi-supervised learning when applied to extremely low-resource ASR tasks.

## 1.2 Transfer learning

For pretrained large models with fine-tuning, wav2vec2 (Baevski et al., 2020; Conneau et al., 2020) is one of the leading representative models. Wav2vec2 has shown strong transfer learning capabilities on ASR tasks. Using models like wav2vec2, the CER can be significantly reduced (Yi et al., 2021). Although transfer learning is generally effective, its performance can be significantly constrained, if there is a large gap between the target and pretrained speech domain. For example, the wav2vec2 model (Baevski et al., 2020) is pretrained on English audio data in the English speech domain, which gives a 4.8% CER by using 10 minutes of labeled English audio data to fine-tune. However, it only achieves a 28.32% CER even using 27k training utts in the Japanese domain (Yi et al., 2021).

## 2 Related work

There have been many attempts to use TTS for data augmentation to benefit ASR tasks, e.g., Laptev et al. (2020); Rossenbach et al. (2020); Sun et al. (2020); Li et al. (2018); Ueno et al. (2021); Rosenberg et al. (2019); Tjandra et al. (2017). Among them, Ueno et al. (2021) focuses on the representation aspect. Results in Rosenberg et al. (2019) indicate the effectiveness of TTS data enhancement, although it may be not as good as the model trained on real speech data. Tjandra et al. (2017) takes advantage of the close connection between TTS and ASR models.

The idea of leveraging concatenative synthesis text-to-speech system to boost ASR is also explored in Du et al. (2021); Zhao et al. (2021). However, in the audio splicing data augmentation method developed in Du et al. (2021), they just replace the English audio part of the code-switching audio, which is a simple and preliminary splicing method. Therefore, the spliced audio diversity is limited, and it is difficult to introduce the audio containing OOV (out of vocabulary) texts. Zhao et al. (2021) focused on the adaptation to new domains and is out of the scope of low-resource tasks.

It is also worth mentioning that a recent work Min et al. (2022) shows that competitive performance on Mandarin ASR tasks can be achieved with only 10 hours of Mandarin audio data using a novel concatenative synthesis text-to-speech system. The process mainly involves training models on one Mandarin audio dataset, mapping characters to pinyin using a character-pinyin dictionary, and synthesizing audio by concatenating pinyin-audio pairs. This method owns many properties that are well-suited for low-resource ASR tasks, as it does not require additional labeled audio data, hence it is efficient, interpretable, and convenient for human intervention. For adaption, a simple energy normalization is provided in Min et al. (2022) instead of using other complex energy normalization methods such as Lostanlen et al. (2018).

# 3 Method

Both semi-supervised learning and transfer learning methods face challenges in low-resource speech recognition tasks. Besides, training a reliable speech synthesis system in such settings is essentially quite difficult. As a solution, our approach uses speech concatenation synthesis as a TTS data augmentation technique. Specifically, for each text to be synthesized, we find the corresponding audio for each word in the text, then normalize their energy and perform concatenation. The key step here is to find the audio for each word in texts, which is accomplished via forced alignment on labeled data. To overcome any possible out-of-vocabulary (OOV) issues, we introduce the concept of meta-audio.

Our approach has two benefits. First, it provides a manner to mix labeled audios in the time domain, and fully utilizes labeled audios to help the model learn more robust acoustic features. Second, it easily allows the model to learn other textual information, benefiting the model to make accurate predictions.

To describe the overall framework of MAC, we use a Bayesian sampling framework. In Section 3.1, we briefly introduce the symbols and notations used, and also the general data configuration for low-resource speech recognition. Sections 3.2 and Section 3.3 introduce the notion of meta-audios and demonstrate its bridging role in the speech concatenation process. Sections 3.4 and Section 3.5 describe the position of forced alignment in the Bayesian sampling process from a mathematical perspective. Section 3.6 presents the benefits of energy normalization in improving the quality of synthesized audios. Finally, Section 3.7 summarizes the overall procedure and gives an algorithmic characterization of the whole synthesis process.

## 3.1 General audio datasets

The mathematical formulation is as follows. Denote the audio wave space by $\mathcal{X}$, and the transcription text space by $\mathcal{Y}$, i.e., $\mathcal{X} = \{x : \text{all the audio waves}\}$, $\mathcal{Y} = \{y : \text{all the transcription texts}\}$. In general, for ASR tasks, the labeled audio dataset consists of audio-transcription pairs $\{(x_i, y_i)\}_{i=1}^N$ sampled from a certain underlying distribution $P$, and $\{x_i\}_{i=1}^N \sim P_x$, $\{y_i\}_{i=1}^N \sim P_y$ with $P_x$, $P_y$ as the marginal distribution of $P$.

Obtaining a practical speech recognition model often requires hundreds or even thousands of hours of audio-transcription pairs for training. Unfortunately, getting audio-transcription pairs is usually expensive. In fact, in many scenarios such as dialects, one can only access around ten hours of audio-transcription pairs. However, the audio-only data $x \sim P_x$ and text-only data $y \sim P_y$ are respectively easier to access. Formally, we have a paired dataset $\mathcal{D} = \{(x_i, y_i)\}_{i=1}^N$, an audio-only dataset $\mathcal{D}_{\text{audio}} = \{x_i\}_{i=1}^{N_1}$ and a text-only dataset $\mathcal{D}_{\text{text}} = \{y_i\}_{i=1}^{N_2}$ with $N_2 \gg N_1 \gg N$. This is the typical setting of low-resource ASR tasks. The goal is to sample paired audio-transcription data $(x, y)$ from the underlying distribution $P$.

Basically, we have

$$P(x, y) = P_y(y)P(x \mid y), \tag{1}$$

where $P_y(y)$ corresponds to the distribution of transcriptions, and $P(x \mid y)$ denotes the distribution of audios conditioned on a certain transcription $y$. Therefore, the sampling of new data (audio-transcription pair) $(x, y)$ can be divided into two stages. First, the transcription text $y$ is sampled from $P_y(y)$, and then the audio $x$ corresponding to the previous transcription text $y$ is sampled from $P(x \mid y)$. The first stage, i.e., sampling of the transcription text $y$, is relatively easy since there is usually sufficient text-only data in $\mathcal{D}_{\text{text}} = \{y_i\}_{i=1}^{N_2}$, We will simply count the frequency of each transcription text in the text-only dataset $\mathcal{D}_{\text{text}}$, and use the frequency to estimate the corresponding probability. That is, $P_y(y) \approx \tilde{P}_y(y) = \frac{1}{N_2} \sum_{y_i \in \mathcal{D}_{\text{text}}} \delta(y_i)$, where $\delta(y_i)$ denotes the Dirac function that is equal to 1 if $y = y_i$, and 0 for any $y \neq y_i$, and recall that $N_2$ is the total number of transcription texts in $\mathcal{D}_{\text{text}}$. Therefore, the key is to analyze and maximize $P(x \mid y)$, which is in fact a TTS task.

## 3.2 Meta-audio set and meta-audio sequence space

In order to perform further "decoupling" analysis on the conditional probability distribution $P(x \mid y)$, we first introduce the meta-audio set, represented by $\mathcal{A}$. Here, meta-audios refer to the basic modeling units of specific language pronunciations. For example, there are about 50 phonemes in English. If we want to use these phonemes as modeling units to characterize the English pronunciation, these phonemes form a

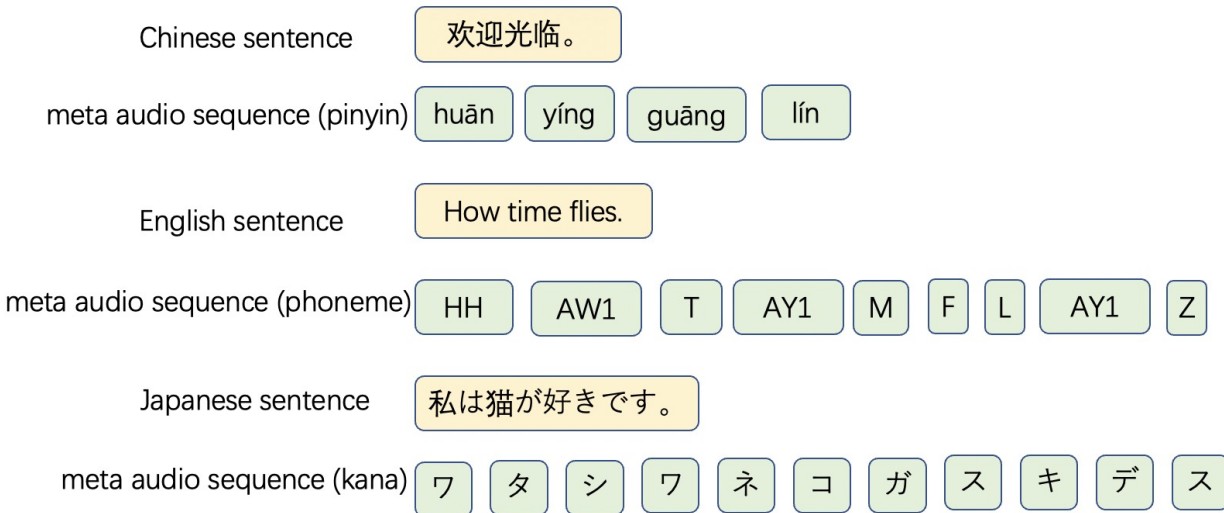

Figure 1: Examples of the mapping function **t** in Chinese, English and Japanese. The function $\mathbf{t} : \mathcal{Y} \to \mathcal{A}$ maps a transcription text to its corresponding meta-audio sequence which reflects language-specific pronunciation rules. Hence, **t** can act as a fusion of prior knowledge of pronunciation rules.

nature meta-audio set of English. In our framework, we have designed the selection of meta-audio sets to be flexible, taking into account the unique characteristics of different languages. For example, in English, a meta-audio set can be created by using phonemes directly, or by fusing multiple phonemes together as a single unit. Similarly, for Mandarin, meta-audio sets can be determined based on pinyin, either in a tone-sensitive or tone-insensitive manner, depending on the requirements of the specific application. For Japanese, kana can be used as meta-audio sets. This flexibility allows us to adapt the meta-audio sets to the specific language being considered, making our approach versatile and adaptable to different linguistic contexts. By considering different options for meta-audio set creation, we can optimize our approach for each language, ensuring accurate and effective results in our audio processing tasks.

**Mapping function** $\mathbf{t} : \mathcal{Y} \to \mathcal{A}$. To reflect language-specific pronunciation rules, we also need a function $\mathbf{t} : \mathcal{Y} \to \mathcal{A}$ to map a transcription text to its corresponding meta-audio sequence. The construction of **t** requires prior knowledge of pronunciation rules. Obviously, different languages may have different mappings, and even different meta-audio sequences in the same language may have different mappings. Figure 1 shows examples of **t** for Chinese, English and Japanese.

### 3.3 Decoupling analysis

In this section, we perform a fine-grained decoupling analysis on the probability $P(x \,|\, y)$, i.e., the conditional distribution of audios given transcriptions. On the one hand, when an audio $x \in \mathcal{X}$ is given, we have a conditional distribution $P(y \,|\, x)$. This is in fact the goal of ASR tasks: predict corresponding texts given audios by estimating $P(y \,|\, x)$. On the other hand, when a transcription $y \in \mathcal{Y}$ is given, we are supposed to model the desired conditional distribution $P(x \,|\, y)$. One can decompose $P(x \,|\, y)$ via the meta-audio sequence space $\mathcal{A}$ and the mapping function **t** as follows:

$$
\begin{aligned}
P(x \,|\, y) &= \sum_{a \in \mathcal{A}} P(x, a \,|\, y) \\
&= \sum_{a \in \mathcal{A}} P(x \,|\, a, y) P(a \,|\, y) \\
&= P(x \,|\, a = \mathbf{t}(y), y).
\end{aligned}
\tag{2}
$$

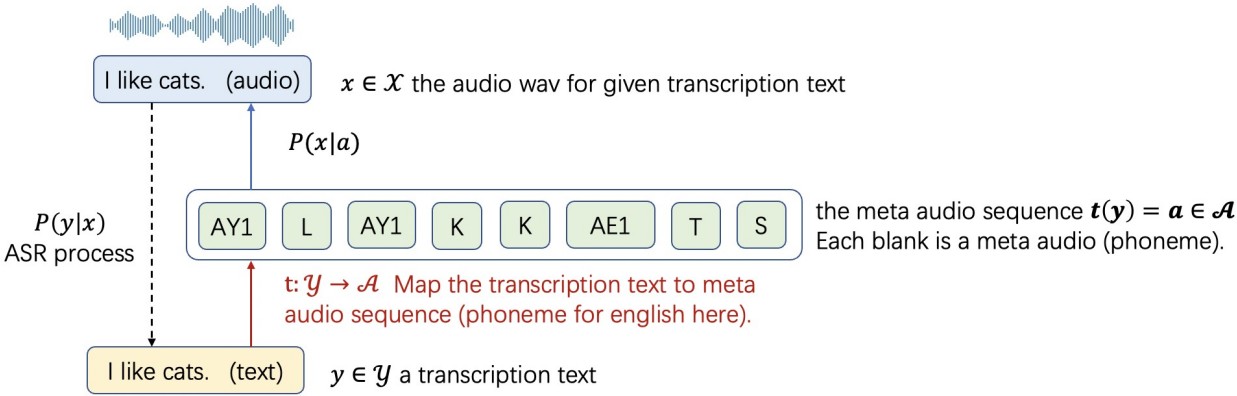

Figure 2: An illustration of the decoupling process. The goal of ASR tasks is to estimate $P(y\,|\,x)$ and the goal of TTS tasks is to estimate $P(x\,|\,y)$. In order to simplify the modeling of $P(x\,|\,y)$, we introduce the mapping function $\mathbf{t}$ to transform $P(x\,|\,y)$ into $P(x\,|\,a)$. The dimension of $a$ is generally much smaller than $y$.

Here, we applied the fact that $P(a\,|\,y)$ is a degenerate distribution with the probability 1 at $a = \mathbf{t}(y)$. Furthermore, the meta-audio sequence contains all the pronunciation information of transcription texts, hence

$$P(x\,|\,a = \mathbf{t}(y), y) = P(x\,|\,a = \mathbf{t}(y)), \tag{3}$$

which gives

$$P(x\,|\,y) = P(x\,|\,a = \mathbf{t}(y)). \tag{4}$$

Figure 2 illustrates this decoupling process.

### 3.4 Bayesian inference on $P(x\,|\,a)$

Based on Section 3.3 and Eq. (4), instead of analyzing $P(x\,|\,y)$ directly, we turn to study $P(x\,|\,a)$. This is usually much easier, since $a \in \mathcal{A}$ often has a *much lower dimension* than $y \in \mathcal{Y}$. Here, the dimension refers to the number of element classes per position of $a \in \mathcal{A}$ or $y \in \mathcal{Y}$. For example, in English, the dimension of $a \in \mathcal{A}$ is about 50 (here we naturally select the phonemes as meta-audios for simple interpretation), while the dimension of $y \in \mathcal{Y}$ could be much higher since there is a large number of English words.

Notice that

$$P(x\,|\,a) \propto P_x(x)P(a\,|\,x), \tag{5}$$

the goal is now converted to maximize $P_x(x)$ and $P(a\,|\,x)$ in order to maximize $P(x\,|\,a)$. Recall that $P_x(x)$ denotes the prior probability of audios, we will discuss it later (in Section 3.6). For $P(a = (a^{(1)}, a^{(2)}, ..., a^{(n)})\,|\,x)$, we have

$$P(a\,|\,x) = \sum_{\mathbf{s}} \prod_{i=1}^{n} P\left(a^{(i)}\,|\,x^{(i)} = [x_{s_i}, x_{s_{i+1}}]\right) \tag{6}$$

Here, $\mathbf{s} = (s_1, s_2, ..., s_{n+1})$ represents the time slice of $x \in \mathcal{X}$. Eq. (6) holds because: 1) the audio wave $x$ can be also properly divided (maybe not unique) to obtain the clip $x^{(i)} = [x_{s_i}, x_{s_{i+1}})$ corresponding to each $a^{(i)}$. For instance, we can segment an English audio wave of one sentence and get the audio wave segmentation corresponding to the sentence's meta-audio sequence;[1] 2) the audio wave clip $x^{(i)}$ in $x$ is monotonous with respect to $a^{(i)}$ in $a$. That is, the timestamps of audio waves and meta-audios must match with each other, i.e., $x^{(i)}$ and only $x^{(i)}$ corresponds to $a^{(i)}$; 3) for simplicity, we treat this correspondence independently.

---

[1]Here, the meta-audio sequence is the phoneme sequence, since we naturally select phonemes as meta-audios for simple interpretation.

Unfortunately, it is still quite expensive to consider all the time slices $\mathbf{s} = (s_1, s_2, ..., s_{n+1})$ in Eq. (6). However, for each time-fixed slice $\mathbf{s}^0 = (s_1^0, s_2^0, ..., s_{n+1}^0)$, we can derive a lower bound of $P(a \,|\, x)$:

$$
\begin{aligned}
P(a \,|\, x) &= \sum_{\mathbf{s}} \prod_{i=1}^{n} P\left(a^{(i)} \,|\, x^{(i)} = [x_{s_i}, x_{s_{i+1}}]\right) \\
&\geq \prod_{i=1}^{n} P\left(a^{(i)} \,|\, x^{(i)} = [x_{s_i^0}, x_{s_{i+1}^0}]\right).
\end{aligned}
\tag{7}
$$

### 3.5 Optimization of $P(a \,|\, x)$

According to Eq. (7) in Section 3.4, we can approximately maximize $P(a = (a^{(1)}, a^{(2)}, ..., a^{(n)}) \,|\, x)$ by maximizing a lower bound determined by a fixed partition. The right hand side of Eq. (7) can be maximized by performing force alignment (Kim et al., 2021; López & Luque, 2022). Specifically, we first map the transcription text $y \in \mathcal{Y}$ in the labeled audio dataset $\mathcal{D} = \{(x_i, y_i)\}_{i=1}^{N}$ into $a \in \mathcal{A}$ to get a corresponding dataset $\{(x_i, a_i)\}_{i=1}^{N}$. Then, we train the ASR model and perform force alignment on $\{(x_i, a_i)\}_{i=1}^{N}$ to get the audio wave clip corresponding to the meta-audio element $a_i$ (with high probability). For further efficiency, we can store the forced alignment results and build a database $\mathcal{B}$, whose procedure is illustrated in Figure 3. When we synthesize audios from texts, we first convert the text $y$ to the meta-audio $a$, then query the audio clip corresponding to each meta-audio element. If there are more than one answers in the database, we just randomly select one. The time slice $\mathbf{s}^0$ in Eq. (7) is implicitly considered when we concatenate these audio clips to form a complete audio wave, since when we concatenate the audio clips corresponding to each meta-audio element in the sequence, it automatically forms a time slice $\mathbf{s}^0$.

**Remark 1** *(**Database size**) For each element $a^{(i)}$, we may get different corresponding audio clips (with high probability) by performing force alignment on $\{(x_i, a_i)\}_{i=1}^{N}$. We just store all of them in the database $\mathcal{B}$ to enrich the selections and increase the diversity of synthesized audios.*

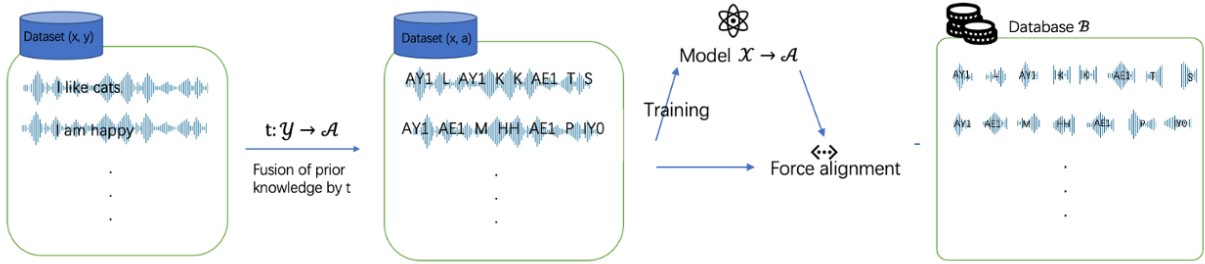

Figure 3: The process of building the database $\mathcal{B}$. Note that we can get audio clips from the training set, hence we can train an ASR model and use it for the forced alignment on the same training set, which is very helpful in extremely low-resource conditions to obtain high quality audio clips.

### 3.6 Energy normalization

In this section, we discuss energy normalization and its benefits. There are many energy normalization methods, and here we follow the operation described in Min et al. (2022). Specifically, we average the energy of sampled audio clips $(x^{(1)}, x^{(2)}, \ldots, x^{(n)})$ corresponding to the meta-audio sequence $a = (a^{(1)}, a^{(2)}, \ldots, a^{(n)}) \in \mathcal{A}$. That is,

$$
E = \frac{\sum_{i=1}^{n} \|x^{(i)}\|}{n},
\tag{8}
$$

$$
\left\{x^{(i)}\right\}_{i=1}^{n} \rightarrow \left\{\frac{x^{(i)}}{\|x^{(i)}\|} * E\right\}_{i=1}^{n}.
\tag{9}
$$

The reason for energy normalization is that the audio wave obtained by combining these audio clips as presented in Section 3.5 may give a high probability $\prod_{i=1}^{n} P\left(a^{(i)}|x^{(i)}\right)$ but do not take the term $P_x(x = (x^{(1)}, x^{(2)}, \ldots, x^{(n)}))$ in Eq. (5) into account. For example, the volume of the generated $(x^{(1)}, x^{(2)}, \ldots, x^{(n)})$ may change rapidly and frequently, leading to a small $P_x(x = (x^{(1)}, x^{(2)}, \ldots, x^{(n)}))$. Mathematically, this can be understood as follows: the support set of $P_x$ is likely to be a very small (proper) subset in the whole audio space $\mathcal{X}$. Therefore, simply merging these audio clips may cause the synthesized audio wave $x = (x^{(1)}, x^{(2)}, \ldots, x^{(n)})$ corresponding to $a = (a^{(1)}, a^{(2)}, \ldots, a^{(n)})$ to be severely distorted.

There are two benefits of this operation. First, this normalization technique helps to mitigate the rapid and frequent changes in volume, resulting in a more stable and natural synthesis of the audio wave $x = (x^{(1)}, x^{(2)}, \ldots, x^{(n)})$. This distortion issue is solved by incorporating energy normalization, which ensures that the generated audio clips are combined in a way that maintains their original characteristics and preserves the overall quality of the synthesized audio wave, thereby improving the quality of the synthesized audio output.

Second, the energy normalization in Eq. (8) and (9) will not affect $\prod_{i=1}^{n} P\left(a^{(i)}|x^{(i)}\right)$ too much since it only involves linear scaling of $x^{(i)}, i = 1, 2, \ldots, n$. We further have:

$$P\left(a^{(i)}|x^{(i)}\right) \approx P\left(a^{(i)}|\frac{x^{(i)}}{\|x^{(i)}\|} * E\right). \tag{10}$$

In a nutshell, by incorporating energy normalization, we ensure that the generated audio clips are combined in a way that maintains their original characteristics and preserves the overall quality of the synthesized audio wave, thereby improving the quality of the synthesized audio output.

Certainly, other (more complex) methods are also widely used in concatenative synthesis-based text-to-speech (TTS) systems (Tabet & Boughazi, 2011; Khan & Chitode, 2016). In the following experiments, energy normalization as described in Eq. (8) and (9) is easy to implement and works well. More importantly, our ultimate goal is to handle ASR tasks, hence it is unnecessary to pay too much attention to the fine-grained quality of audios generated by TTS, which may lead to over-complex models. However, improving the quality of synthesized audios may give better results, and we will further discuss the applicability of other (more complex) methods in the future work.

### 3.7 Summary of the MAC framework

Overall, the MAC framework is a method for synthesizing audios. The following algorithm outlines all the steps involved:

---
**Algorithm 1** MAC
---
**Input:** Labeled audio dataset $\mathcal{D} = \{(x_i, y_i)\}_{i=1}^{N}$, text-only dataset $\mathcal{D}_{\text{text}} = \{y_i\}_{i=1}^{N_2}$
 1: Determine a proper meta-audio set $\mathcal{A}$ according to the language pronunciation rules
 2: Create the mapping function $\mathbf{t} : \mathcal{Y} \rightarrow \mathcal{A}$ as pronunciation rules.
 3: Map $\mathcal{D} = \{(x_i, y_i)\}_{i=1}^{N}$ into $\{(x_i, a_i)\}_{i=1}^{N}$ using $\mathbf{t}$
 4: Perform force alignment on $\{(x_i, a_i)\}_{i=1}^{N}$, then build the database $\mathcal{B}$
 5: Sample the text $y \in \mathcal{Y}$ from $\tilde{P}_y(y) = \frac{1}{N_2} \sum_{y_i \in \mathcal{D}_{\text{text}}} \delta(y_i)$ and get $a = (a^{(1)}, a^{(2)}, ..., a^{(n)}) = \mathbf{t}(y)$
 6: Randomly select one audio clip $x^{(i)}$ in $\mathcal{B}$ for each $a^{(i)}$ in the meta-audio sequence $a = (a^{(1)}, a^{(2)}, ..., a^{(n)})$
 7: Perform regularization energy normalization and concatenate the audio clips
 8: Obtain the synthesized audio-transcription pair data $(x, y)$, where $x$ is from Step 7 and $y$ is from Step 5
**Output:** The synthesized audio-transcription pairs.
---

## 4 Experiments

We verify the effectiveness of MAC on three real low-resource Cantonese, Taiwanese, and Japanese ASR tasks on the corresponding dataset in the Common Voice datasets.[2] The experiments show that MAC outperforms the large-scale wav2vec2 pretrained model with fine-tuning, and achieves very competitive results on these tasks.

### 4.1 Datasets and pre-processing

We use the training split of the corresponding dataset in the Common Voice datasets (11.0 version) as the original labeled data, which is a publicly available multilingual audio dataset contributed by volunteers around the world. Also, we generate synthetic data using the proposed MAC. The detailed number of original labeled data in the training split and synthetic data generated by MAC is reported in Table 2. We use the test split of corresponding datasets to evaluate the performance.

| Dataset | Original | Synthesized by MAC | All |
|---|---|---|---|
| Cantonese | 8,423 | 63,999 | 72,422 |
| Taiwanese | 6,568 | 69,315 | 75,883 |
| Japanese | 6,505 | 33,099 | 39,604 |

Table 2: Detailed number of original labeled audio data in the training split and synthetic data generated by MAC in the Cantonese, Taiwanese and Japanese ASR tasks.

### 4.2 Hybrid CTC/attention architecture

To ensure a comprehensive speech recognition solution, we utilize a cutting-edge hybrid CTC/attention model with the advanced Wenet toolkit (Yao et al., 2021). The encoder architecture is based on the Conformer, which is known for its high performance in speech modeling. Meanwhile, the decoder uses the Transformer decoder, which is another well-established architecture in the field of speech recognition.

The encoder consists of 6 Conformer blocks, each comprising a sequence of multi-head self-attention, CNN and feed-forward modules. The attention layer in encoder has an output dimension of 256 and is equipped with 4 attention heads, the kernel of cnn module is 15 and the feed-forward layer contains 512 units. The decoder consists of 3 blocks of multi-head attention, with 4 attention heads and 512 units in the feed-forward layer .

We select a dropout rate of 0.3, which helps prevent overfitting and enable generalization. We also employ the Swish (Ramachandran et al., 2017) as the activation function.

### 4.3 Setup

We conduct the experiments on 2×RTX 3090GPUs (24GB) and 4×P100GPUs (16GB). We set the maximum epoch as 300, and the model is trained using the Adam optimizer with a learning rate of 0.002. We implement a warm-up learning rate scheduler, and the gradient clip is set as 5. Besides, data augmentation techniques such as speed perturbation and spectral augmentation are applied. We also perform label smoothing with a weight of 0.1 and use a hybrid CTC/attention loss with a weight of 0.3.

For pre-processing, we set the maximum input length to be 40960ms, and the minimum length to be 0. The maximum length and minimum length of tokens is set as 200 and 1, respectively. We use a re-sample rate of 16000, and 80 mel-bins with a frame shift of 10 and a frame length of 25 for feature extraction. Besides, we randomly shuffle and sort the data during training.

---

[2]We use the zh-HK dataset in the Common Voice datasets for the Cantonese ASR task.

| Task | Model | CER |
|---|---|---|
| Cantonese ASR | Hybrid CTC/attention model | 32.5 |
| | Hybrid CTC/attention model + MAC | 12.7 |
| | wav2vec2 + fine-tuning | 15.4 |
| Taiwanese ASR | Hybrid CTC/attention model | 51.3 |
| | Hybrid CTC/attention model + MAC | 22.0 |
| | wav2vec2 + fine-tuning | 18.4 |
| Japanese ASR | Hybrid CTC/attention model | 45.3 |
| | Hybrid CTC/attention model + MAC | 25.0 |
| | wav2vec2 + fine-tuning | 24.9* |

Table 3: CERs for Cantonese, Taiwanese and Japanese on ASR tasks. We use the advanced hybrid CTC/attention model and test with CTC prefix beam search. It is shown that MAC can boost performance significantly, and also achieve competitive results compared to the pretrained wav2vec2 model with fine-tuning.

For Cantonese, Taiwanese, and Japanese ASR tasks, the natural choice of the meta-audio set is based on their respective pronunciation rules. Here, we use Cantonese pinyin for Cantonese, pinyin for Taiwanese, and kana for Japanese. Taking advantage of the flexibility of the meta-audio notion, some adjustments can be made on each language's meta-audio set. For instance, we do not distinguish tones for Cantonese pinyin, but for pinyin, which is used to construct the Taiwanese meta-audio set, we treat the tones respectively. Due to the fact that for a transcription text $y = (y^1, y^2, ..., y^m)$ in Cantonese or Taiwanese, we generally have the approximation

$$\mathbf{t}(y^1, y^2, ..., y^m) \approx (\mathbf{t}(y^1), \mathbf{t}(y^2), ..., \mathbf{t}(y^m)), \tag{11}$$

the order of Step 3 and Step 4 is reversed to avoid additional training of the model for forced alignment[3] and the overall process is simplified. For Japanese, Eq. (11) no longer holds, hence the order of Step 3 and Step 4 is not reversed.

For each language, the mapping function $\mathbf{t}$ is used to map the transcription text $y \in \mathcal{Y}$ to their meta-audio sequence $a \in \mathcal{A}$. The text-only dataset $\mathcal{D}_{\text{text}}$ for Step 5 is obtained by using transcriptions in the validation split of the respective language dataset in the Common Voice datasets, and we remove transcriptions appearing in the test split. For Step 7, the energy normalization method described in Eq. (8) and (9) is applied.

## 4.4 Performance

We evaluate the performance of MAC on the test split of the corresponding language dataset in the Common Voice datasets. The evaluation metric is the character error rate (CER), which measures the difference between the predicted and ground truth transcripts. We compare the results with baselines obtained by training directly on the original training split of labeled data with data augmentation techniques such as speed perturbation and spectral augmentation mentioned above.

The main results are shown in Table 3. We refer to the wav2vec2+fine-tuning results from the URLs,[4] rounding to one decimal place. In all three tasks, the MAC method reduces CERs by more than **15%**. Here, we only show results using the CTC prefix decoding, but in fact, using the attention rescore decoding may yield better results, which can be found in the appendix.

Furthermore, MAC outperforms wav2vec2 (with fine-tuning) and achieves a new state-of-the-art (SOTA) on the Common Voice Cantonese ASR tasks. The asterisk (*) in Table 3 indicates that the 24.9 CER is achieved by using extra data compared to the Japanese audio dataset in the Common Voice datasets during fine-tuning the wav2vec2 model. It is shown that MAC relatively improves the performance by about **20%** on the Cantonese ASR task (and we can achieve the 10.9 CER with attention rescore decoding on the Cantonese

---

[3]We can directly use the baseline model in Table 3 to perform forced alignment.

[4]Cantonese: `https://huggingface.co/ctl/wav2vec2-large-xlsr-cantonese`, Japanese: `https://huggingface.co/qqhann/w2v_hf_jsut_xlsr53`, and Taiwanese: `https://huggingface.co/voidful/wav2vec2-large-xlsr-53-tw-gpt`.

ASR, with an around 30% relative improvement compared to fine-tuning the wav2vec2 model. It is a new SOTA to the best of our knowledge, and more details can be found in the appendix.) Additionally, for the Taiwanese and Japanese ASR tasks, MAC also achieves comparable results to the fine-tuned wav2vec2 model (and we can achieve the 23.4 CER with attention rescore decoding on the Japanese ASR, with an around 6% relative improvement compared to fine-tuning the wav2vec2 model. See more details in the appendix.)

## 5 Ablation studies

In this section, we conduct ablation experiments to evaluate the performance of MAC, aiming to gain a deeper understanding of MAC and its various components, as well as to provide insights into the most effective ways to use this method for ASR modeling.

We focus on three key aspects:

1. What are the advantages of MAC over other NN-based TTS methods? How much room is there to improve the ASR performance by adding synthetic audio?

2. What is the impact of synthetic data quantity on the ASR performance?

3. Does energy normalization really help to improve the ASR performance?

### 5.1 Comparison: MAC and other NN-based methods

| Language | Data | Attention | CTC greedy | CTC prefix | Attention rescore |
|---|---|---|---|---|---|
| Cantonese | Original training split | 44.1 | 32.5 | 32.5 | 30.6 |
| | + Synthesized by MAC | 11.0 | 12.7 | 12.7 | 10.9 |
| | + Validation split | 6.7 | 7.0 | 7.0 | 6.1 |
| Japanese | Original training split | 72.1 | 45.3 | 45.3 | 44.5 |
| | + Synthesized by MAC | 24.3 | 25.1 | 25.0 | 23.4 |
| | + Tacotron2 synthesized data | 25.8 | 24.2 | 24.2 | 23.0 |
| | + Validation split | 8.5 | 8.3 | 8.3 | 7.5 |
| Taiwanese | Original training split | 55.3 | 51.3 | 51.3 | 48.2 |
| | + Synthesized by MAC | 18.6 | 22.0 | 22.0 | 19.5 |
| | + Validation split | 10.0 | 11.5 | 11.5 | 9.9 |

Table 4: Comparison of speech recognition results. The "original training split" refers to only using the training split data from the Common Voice datasets, "+ synthesized by MAC" refers to adding MAC synthesized data, and "+ validation split" refers to adding the validation split data in the Common Voice datasets, "+ Tacotron2 synthesized data" refers to adding the synthesized data generated by the Tacotron2 model (Shen et al., 2018)

.

In this section, we discuss the advantages of MAC over other NN-based TTS systems under low-resource scenarios and report the results of adding real data instead of synthesized data to assess the remaining room for improvement. Table 4 presents the corresponding experimental results.

One significant advantage of MAC is that it can be applied normally compared to other NN-based TTS systems under low-resource conditions. The limited availability of annotated audio data makes it challenging to train an NN-based TTS system for audio data synthesis. In fact, for languages like Cantonese, Taiwanese, and Japanese, we are unable to train an NN-based TTS system successfully using the limited labeled data in the Common Voice datasets.

For comparison, we use the Japanese TTS system,[5] which employs extra data for training. Unfortunately, we find no publicly available TTS systems of Cantonese and Taiwanese, likely due to the scarcity of annotated

---

[5]This is available at `https://github.com/coqui-ai/TTS`.

data. To ensure a fair comparison, we fix the number of synthesized samples as 30,000, and we use traditional data augmentation techniques such as speed perturbation and SpecAugment (Park et al., 2019) in all settings. Nonetheless, MAC achieves comparable results to the NN-based TTS system (Tacotron2), but without the requirement on additional data for training or extensive inference operations, as reported in Table 4.

We further add real data instead of synthesized data to explore the remaining room for improvement. In Table 4, "+ validation split" denotes adding the validation split data in the Common Voice datasets, which can be viewed as an upper bound of the optimal performance. The results demonstrate that adding MAC-synthesized data significantly improves the recognition accuracy compared to only using the original training split data in the Common Voice datasets, leading to a limited improvement possibility.

In summary, the results suggest that adding MAC-synthesized speech data is comparable to adding NN-based TTS synthesized data in improving the ASR performance but without requiring additional data or extensive inference computation. Additionally, the results here imply little room for further improvement by improving the quality of synthesized data using an advanced NN-based TTS system.

### 5.2 Impact of synthetic data quantity

The amount of data for training an ASR model is an important factor that can significantly impact the performance. In this section, we explore the impact of synthetic data quantity on the performance of different ASR models on the Taiwanese and Cantonese datasets. Specifically, we examine how the CER changes when more synthetic data is added to the training set. The results are shown in Table 5.

| Taiwanese | | | | |
|---|---|---|---|---|
| Utt | CER (%) | | | |
| | Attention | CTC greedy | CTC prefix | Attention rescore |
| 10000 | 25.3 | 30.0 | 29.9 | 26.8 |
| 30000 | 18.6 | 23.3 | 23.3 | 20.7 |
| All (69315) | 18.6 | 22.0 | 22.0 | 19.5 |
| Cantonese | | | | |
| Utt | CER (%) | | | |
| | Attention | CTC greedy | CTC prefix | Attention rescore |
| 10000 | 19.9 | 19.5 | 19.5 | 17.4 |
| 30000 | 12.8 | 14.1 | 14.1 | 12.0 |
| All (63999) | 11.0 | 12.7 | 12.7 | 10.9 |

Table 5: The effect of synthetic data quantity on the ASR performance.

Based on Table 5, we can observe that there is a decrease in the CER for all models when more data is added, but the magnitude becomes smaller as the amount of synthetic data increases. For example, on Taiwanese, the model with the CTC prefix decoding mode has a CER of 29.9% with 10,000 utterances, which drops to 23.3% with 30,000 utterances, but only drops to 22.0% with all 69,315 utterances. Similarly, on Cantonese, the model with the CTC prefix decoding mode has a CER of 19.5% with 10,000 utterances, which drops to 14.1% with 30,000 utterances, but only drops to 12.7% with all 63,999 utterances. It indicates that adding more synthetic data does help to improve the performance, but the improvement becomes smaller as the amount of data increases.

The results suggest that adding more data can continue to improve the performance, but it may be not practical or feasible to collect and use all the data. The reasonable amount of required data depends on the desired level of performance and the data availability. It is necessary to trade off between the cost of collecting data and the potential improvement in performance.

### 5.3 Effect of energy normalization

In this section, we explore the role of energy normalization. Specifically, we present experimental results with and without energy normalization on two different languages. The results demonstrate that energy normalization can improve the quality of synthesized audios, resulting in lower error rates.

| Language | Model | Attention | CTC greedy | CTC prefix | Attention rescore |
|----------|-------|-----------|------------|------------|-------------------|
| Cantonese | Without normalization | 12.5 | 13.8 | 13.8 | 11.8 |
| | With normalization | 11.0 | 12.7 | 12.7 | 10.9 |
| Taiwanese | Without normalization | 18.6 | 22.3 | 22.3 | 20.2 |
| | With normalization | 18.6 | 22.0 | 22.0 | 19.5 |

Table 6: Comparison of speech recognition models on Cantonese and Taiwanese with and without energy normalization.

Table 6 presents results for four different models (Attention, CTC greedy, CTC prefix, and attention rescore) on two different languages (Cantonese and Taiwanese) with and without energy normalization. It is shown that energy normalization generally improves the performance of corresponding models, leading to lower character error rates, which demonstrates the importance of energy normalization. In general, the use of energy normalization can enhance the quality of synthesized audios, which potentially leads to a better ASR modeling.

## 6 Conclusion

In this work, we propose the MAC framework as a unified solution for low-resource automatic speech recognition tasks. The framework incorporates a broad notion of meta-audio sets that enables its application as long as there is knowledge of the required pronunciation rules to construct a suitable meta-audio set. Additionally, we provide a clear mathematical description of the MAC framework from the perspective of Bayesian sampling.

Our experiments demonstrate the effectiveness of MAC in low-resource speech recognition tasks, achieving remarkable improvements in accuracy even without a careful tuning of hyper-parameters. Furthermore, the proposed method significantly improves the performance of speech recognition systems in low-resource settings. Our ablation experiments provide insights into the contribution of different components, demonstrating that speech concatenation synthesis with forced alignment, meta-audios, and energy normalization can be a useful data augmentation technique. The MAC method also has some limitations. For example, it requires in-domain texts and prior knowledge of pronunciation rules to construct the meta-audio sets, which may be not always readily available.

To conclude, the present work provides a comprehensive framework that can enhance the performance of speech recognition systems in low-resource settings. In future work, we plan to explore efficient ways to construct the meta-audio sets and combine with other sampling procedures such as Li et al. (2019). We hope that the MAC framework can contribute to the development of low-resource audio recognition.

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

# A   More experimental results

As a supplement of Table 3, we consider about several different decoding modes: CTC greedy search, CTC prefix beam search, attention as well as attention rescore. The complete results are shown in Table 7.

| task | model | decode mode | CER |
|------|-------|-------------|-----|
| Cantonese ASR | Hybrid CTC/attention model | CTC greedy search | 32.5 |
| | | CTC prefix beam search | 32.5 |
| | | attention | 44.1 |
| | | attention rescore | 30.6 |
| | Hybrid CTC/attention model + MAC | CTC greedy search | 12.7 |
| | | CTC prefix beam search | 12.7 |
| | | attention | 11.0 |
| | | attention rescore | **10.9** |
| | wav2vec2 + fine-tuning | - | 15.4 |
| Taiwanese ASR | Hybrid CTC/attention model | CTC greedy search | 51.3 |
| | | CTC prefix beam search | 51.3 |
| | | attention | 55.3 |
| | | attention rescore | 48.2 |
| | Hybrid CTC/attention model + MAC | CTC greedy search | 22.0 |
| | | CTC prefix beam search | 22.0 |
| | | attention | 18.6 |
| | | attention rescore | 19.5 |
| | wav2vec2 + fine-tuning | - | **18.4** |
| Japanese ASR | Hybrid CTC/attention model | CTC greedy search | 45.3 |
| | | CTC prefix beam search | 45.3 |
| | | attention | 72.1 |
| | | attention rescore | 44.5 |
| | Hybrid CTC/attention model + MAC | CTC greedy search | 25.1 |
| | | CTC prefix beam search | 25.0 |
| | | attention | 24.3 |
| | | attention rescore | **23.4** |
| | wav2vec2 + fine-tuning | - | 24.9* |

Table 7: CERs for Cantonese, Taiwanese and Japanese on ASR tasks. We use the advanced hybrid CTC/attention model as the baseline (tested for four decoding modes: CTC greedy search, CTC prefix beam search, attention, and attention rescore). It is shown that MAC can boost performance significantly, and also achieve competitive results compared to fine-tuning the pretrained wav2vec2 model.

