# OpenReview forum: "MAC: A unified framework boosting low resource automatic speech recognition"
_TMLR — Rejected by TMLR_

### Review · Reviewer_cbZY · 2023-03-22

**Summary Of Contributions:**

The key contribution of this submission is a generalisation of previously proposed data augmentation scheme for speech recognition. The scheme belongs to the family of speech synthesis approaches. In particular, it adopts the concatenative speech synthesis framework and makes a certain modifications to concatenated speech pieces to normalise their energy, which also appears to be published before. The submission evaluates the proposed approach on the CommonVoice speech dataset. Three subsets of the CommonVoice dataset have been explored: Cantonese, Taiwanese, and Japanese speech. Compared to not using any form of data augmentation, the proposed approach yields gains.

**Audience:**

No

**Claims And Evidence:**

No

**Requested Changes:**

Please see the weaknesses above. Please also produce an unambiguous description of the algorithm used to generated artificial data (Figure 4 is not adequate as it includes ambiguities like "search the audio clips"). Addressing them is critical for securing a recommendation.

Please reconsider the term meta audio as it is highly confusing. Please improve the quality of your writing (technical presentation, peculiar sentences ("keep their eyes on"), formatting (Relate work, bayesian), use of space (e.g. Figure 5)).

**Strengths And Weaknesses:**

The key strengths of this submission are experiments conducted on 3 non-English subsets of CommonVoice dataset which show that the previously proposed approach may at times be more effective than finetuning of a well known speech recognition system pretrained on English data.

However, there are a number of weaknesses.

First is the limited scope of claimed generalisation of the previously proposed method. The generalisation is mostly verbal and even if experimental results provide some experimental evidence this is hardly possible to infer from the text.

Second is the over-theorising used to describe common steps in the concatenative speech synthesis. Furthermore, this theorising is done in a technically weak fashion. For example, you are making assumptions that are not necessary true in many relevant cases (eq 2). Another example is that you implicitly link choosing one specific segmentation of speech into units with Bayesian inference. Another example is your use of an arbitrary alignment process as the maximisation of P(a|x). Final example, is the crude energy normalisation you are using for creating speech with more consistent energy levels.

Third is the lack of any other data augmentation baseline making it impossible to compare the benefit of this approach. A related issue is not providing any information regarding the state-of-the-art in these subsets.

Fourth is a rather repetitive writing in the main body of the paper as well as in the experimental part. For example, sections 4.3.1, 4.3.2 and 4.3.3 are your opportunity to showcase claimed generalisation but you writing appears to look like a copy-paste.

---

### Review · Reviewer_gdD5 · 2023-03-23

**Summary Of Contributions:**

The paper presents an approach to improving low resource speech recognition by performing concatenative synthesis on the training data, creating a "meta" training set by "audio concatenation" (MAC). Concatenation is done at the phoneme level, so the approach requires additional text data and a lexicon in order to work. Volume normalization is proposed to improve the quality of the synthesis. The authors describe their approach, provide a discussion of related work, and present results on Cantonese, Taiwanese, and Japanese data sets (common voice), evaluated with CER, presumably to factor out the influence of the language model. The approach seems technically straightforward, and the authors report results that are at or near the state of the art for the dataset.


**Audience:**

Yes

**Claims And Evidence:**

No

**Requested Changes:**

Abstract [optional]
- In general, I recommend proofreading of the paper and focusing the paper and its part (some information is repeated in multiple places)

Introduction [optional]
- Please discuss Table 1 in more detail - how does the quality of the initial model affect the quality of the final outcome in the ? Does your approach have a similar dependence on the quality of the initial model (used for segmentation), or not? If not, maybe Table 1 is not needed (or maybe it could be replaced with results on Common Voice)?

Related Work ("Related", not "Relate") [optional]
- It seems Xinjiang Li's work on corpus relatedness sampling (https://arxiv.org/abs/1908.01060) would be useful as a different criterion to use during sampling
- It would be good to clearly distinguish between "approaches" (e.g. HMM-GMM) and "toolkits" (e.g. Kaldi, EspNet) which implement approaches, and cite them accordingly

Methods [addressing these questions is mandatory]
- "bayesian" -> "Bayesian"
- "low recourse" -> "low resource"
- 3.1: why do you formulate the problem in "wave" space, rather than in feature/ vector space? does it not make more sense to interpret the "x" to be Mel spectrograms, or whatever the pre-processing you are using (please describe)
- the consequence is that volume normalization may or may not make a difference any more, e.g. if you use spectrograms and neglect the 0 component (not an unreasonable choice), volume normalization would not make any difference
- please explain how volume normalization can be seen as a simple form of regularization, and ablate the impact on the final results in terms of CER. Does it make any difference?
- 3.2: I find this discussion very generic and think it could be shortened. I can understand that merging different phonemes can generate different "meta audio spaces" - but have you conducted corresponding experiments? Does it matter? If not, may be better to leave out, or only briefly mention it
- 3.3: how are you performing "decoupling analysis" by simply mapping text to pronunciations (e.g. collapsing homophones)? please elaborate, and maybe give examples that relate to English and the languages that use present experiments on
- 3.4/ 3.5: Are Eq 6 and 7 not just a complicated way of describing a Viterbi alignment of the phone sequence to the training data? If so, are you using monophones, or are you using tri-phones or more complicated models?
- 3.6: Here, you are saying that it is possible that a sequence of best segments is not the best overall sequence, right? I.e. the best "a^(1)" and the best "a^(2)" don't form the best "a^(1) a^(2)" together. This is correct, but can you explain how some regularization solves this problem? Maybe "regularization" can improve the score of a given combination, but the fundamental problem is that you seem to be picking the globally best a^(i) and since I don't see any length normalization, you would almost always pick the shortest a^(i), and this seems to be a global pick -- in short: if the target language has 26 phones, the proposed algorithm picks the 26 shortest occurrences of these phones on the original training corpus, and then it concatenates them to create additional training data to generate the "meta" corpus. The volume normalization (performed in time domain) depends on the target utterance and "smooth" the phones somewhat, but essentially, the new data will only be generated from these 26 prototypes - is that correct? If not, please clarify and describe the training algorithm - I see some discussion of the length of the segments in section 3.5 (and storing all of them in the database to enrich the selection, but I honestly do not understand the sampling process, if any)
- Optionally, it may be a good idea to present a few examples (Appendix?), from which the reader can see how the proposed approach works intuitively

Experiments [addressing these questions is mandatory]
- 4.1: please provide a more complete set of baseline results, since common voice is widely used: (1) the overall best performance with only in-domain and with external data, (2) consistent results with e.g. wav2vec, and/ or other more related tools - it would be helpful to better understand how the results achieved here compare to the state of the art and other methods
- 4.1: please motivate the choice of language(s) - why did you pick Cantonese, Taiwanese, Japanese? does the method work in other languages, too?
- 4.1: why not replace Figure 5 by a table and report the details in terms of duration, number of samples, etc?
- 4.1: why do you report 4 different results for each method (greedy, hybrid, ...) - none of these relate to your proposed approach, and they don't seem to add any information (other than "attention" being really bad in Japanese without MAC)?
- 4.1/ 4.2: can you clarify what the "models" are that you compare (parameter size, training strategy, etc), and describe exactly how they are trained (recipes, what was used for training/ validation), on how much data (which splits, ideally by pointing to published recipes), and for how long (early stopping?), etc? I assume that you always (for every language) use the transcriptions of the validation split to generate the meta training corpus (and remove the transcriptions of the test split)? In common voice, given the scripted nature of the corpus, how does that work?
- 4.3: can you also report the corresponding validation CERs, not just the evaluation CERs? ideally also for the baseline/ comparable models, not just for the proposed models? I think it would be helpful to understand where the gains are coming from
- 4.3: in all three languages, you reverse the order of steps 3 and 4 -- why not present it that way, and always reverse them?
- 4.3: can you provide the following ablations:
  - using varying amounts of additional data? have you reached saturation? how much is needed?
  - with and without energy normalization
  - with different types of sampling the training text? how much variety is needed? if training transcriptions don't appear in test, can you still compute perplexities to show what is being synthesized and what is being used?
  - maybe start with different starting models? how much does it affect the quality of the final result?
- 4.3: for Japanese, please present other results that do not use extra data (extra data seems to degrade results, rather than improve them, since there is a domain match between common voice and other domains)
- 4.3: is there any information on speaker overlap between training and testing? given that the additional audio data seems to come from the original training data, this is probably not an issues, but would be interesting to evaluate

Conclusion [optional]
- it would be nice to discuss the limits of this work (e.g. the prompt design of common voice) and explain how this informs future work


**Strengths And Weaknesses:**

Strengths

- The approach is technically straightforward and seems computationally inexpensive, the main cost is in training an initial model, aligning the extra training data, creating the additional training data and training the final model.
- The idea is applicable to a number of situations. Many languages have limited audio data, but available text data, and lexica, so the idea could have significant impact.
- The work could receive significant interest given that this is an active research field, and the results are intriguing.

Weaknesses

- The writing is often vague, and leaves out many technical details (I will ask for clarification below). No details on the amount of additionally generated training data is given (e.g. how many hours, how many samples - only percentages in Table 1)
- The paper does not present any meaningful ablations (e.g. how much does volume normalization help?) and it is therefore unclear where the gains are coming from.
- Even for the presented results, it would be better to show results that vary the training procedure (MAC), rather than the testing procedure (decoding in various ways). The paper does not contain "extensive experiments".
- The paper is very detailed in some aspects, and leaves out other relevant information (see my requested changes below), it would be good to balance the description -- fundamentally, the way the approach is described, I don't see how it generates meaningfully diverse training samples, because based on the description, it should simply select the globally shortest monophones (or syllables) for Cantonese/ Taiwanese/ Japanese, and then concatenate these
- It would be better to use & present a few more baselines, e.g. not just the presented wav2vec baselines, but also the overall best results, and results from closely related methods (e.g. the speech chain, or other concatenative synthesis based approaches?)
- I am not clear if code and data (e.g. the generated text) can be made available so that experiments can be reproduced?

---

### Review · Reviewer_yJR9 · 2023-03-25

**Summary Of Contributions:**

This paper proposes to use concatenation-based speech synthesis as a data augmentation technique for ASR. They use meta information (phoneme, pinyin, kana) and get the alignment between this meta unit and the corresponding speech. They can use text information as a query to create speech and text pair data. The paper also formulates this process with Bayesian sampling. The experiments are conducted with the Common Voice tasks and obtained a reasonable performance improvement from the system without data augmentation by referring to the performance obtained by w2v2 fine-tuning.

**Audience:**

Yes

**Broader Impact Concerns:**

No concern.


**Claims And Evidence:**

Yes

**Requested Changes:**

Experiments
- Please add the other data augmentation techniques and compare the proposed method with them.

Clarity-related requests:
- Abst: "bayesian sampling" --> "Bayesian sampling."
- Abst: "Cantonese ASR task, Taiwanese ASR task, and Japanese ASR task" --> "Cantonese, Taiwanese, and Japanese ASR tasks" I found this issue in the other parts.
- Abst: "common voice datasets" --> "the Common Voice datasets."
- Introduction, 1st paragraph: "Speech-transformer (Dong et al., 2018), Conformer (Gulati et al., 2020), Espnet (Watanabe et al., 2018), Wenet (Yao et al., 2021), LAS (Chan et al., 2015)": This sentence mixes the references from the methods (transformer, conformer, LSA) and the toolkits (Espnet and Wenet). Please separate them.
- Introduction: Some references are not representative ones, e.g., "HMM-GMM (Rodríguez et al., 1997)" and "advanced end-to-end models such as wenet (Yao et al., 2021)." Please find appropriate references, e.g., for HMM-GMM, we can cite more representative papers, including Rabiner, Lawrence R. "A tutorial on hidden Markov models and selected applications in speech recognition." Proceedings of the IEEE 77.2 (1989): 257-286.
- Introduction: It is better to clarify what is the meta audio set more concretely in the Introduction stage.
- Introduction, 3rd paragraph: Please add the baseline numbers and clearly show the improvements.
- Introduction, 3rd paragraph: What are "common speech datasets" and "common spoken?" Are you talking about Common Voice? Please fix them.
- Introduction, 4th paragraph: "Also, semi-supervised learning and transfer learning are currently two main ways to deal with the (labeled) data limit problem." I could not parse this sentence. Please rewrite it.
- Section 1.1, 1st paragraph: "TED" --> "TEDLIUM."
- Table 1: Where can I find the results of the TEDLIUM? Can you add such information to Table 1?
- Section 1.3: These items are mostly listed in Section 1.
- Section 2: Add more detailed information about the energy normalization method with the citations.
- Section 3, "mathematically rigorously from the perspective of bayesian sampling": There are a lot of approximations, and I could not support that this is mathematically rigorous. I still respect your formulation part, so please make this expression milder.
- Section 3.1, 3rd paragraph: "hundreds even thousands of hours" --> "hundreds or even thousands of hours."
- Section 3.1, 4th paragraph: What is $\delta(y_i)$? Please explain it.
- Figure 1: Why have these sentences the same meaning across the language and translated sentences? I think they do not have to be translated sentences for your proposed method.
- Section 3.2: I understand the importance of using the metadata, but it requires linguistic knowledge, which would be challenging for some low-resource languages and also face the OOV issue. It is better to discuss the pros and cons of using metadata.
- Section 3.3: It is better to clarify what is $x$. Is it a log mel filter bank?
- Equation (7): What does ${}^0$ mean? How to obtain it? Please add a concrete explanation about it.
- Section 3.6: I recommend the authors clearly formulate and describe the search issue (how to find $x$). The current description is difficult to follow.
- Section 3.6: I wonder why energy normalization was used to modify $p(a|x)$. The first paragraph in Section 3.6 points out that we should consider $p(x)$, which I agree with based on equation (5). However, the second paragraph does not modify $p(x)$, but the energy normalization was used to modify $p(a|x)$, which confuses me. Please clarify this part.
- Section 4: Please add the version information of the Common Voice database.
- Section 4 "tasks. For instance, we have achieved 10.9% CER on the common voice dataset of Cantonese for ASR tasks, leading an around 30% relative improvement compared to fine-tuning wav2vec2 model.": This sentence already appears in the Introduction, and the authors do not have to describe it again. It looks redundant.
- Table 2: Why did you list the result with various decoding modes? If you want to compare it with the w2v2+fine-tuning purely, you can only list the CTC results.
- Sections 4.3.2 and 4.3.3 are almost the same descriptions as Section 4.3.1. Please remove the redundant explanations.
- Section 4.3.3 "we make some adjustments on the Cantonese pinyin": Why did you use the Cantonese pinyin for Japanese? Is it a typo?


**Strengths And Weaknesses:**

Strengths
- Application of data augmentation based on concatenation-based speech synthesis to multiple languages.
- Formulation based on Bayesian sampling

Weaknesses
- Incremental update from (Min et al., 2022). It is based on concatenation-based speech synthesis and already uses pinyin as metadata.
- No comparisons from the other data augmentation techniques.
- A lot of issues in the clarification (see below).

---

### Decision · Action_Editors · 2023-05-02

**Recommendation:** Reject

**Comment:**

All three reviewers and the action editor thank the authors for the effort they put into revising this submission. It is a much better paper following the revision.

That said, none of the reviewers recommended acceptance of the submission. The primary concerns raised pertain to the clarity and correctness of the paper, which I discuss in detail above.

I have two other recommendations for improving the work.

First, the discussion of prior work is still quite thin, given the large number of papers that have already proposed using TTS methods to synthesize training data for speech recognition systems, and the submission is still missing several papers, including several that focus specifically on low-resource languages:
- Andrea DeMarco, Carlos Mena, Albert Gatt, Claudia Borg, Aiden Williams, Lonneke van der Plas, "Analysis of Data Augmentation Methods for Low-Resource Maltese ASR," arXiv:2111.07793 [cs.CL], https://arxiv.org/abs/2111.07793
- Rodolfo Zevallos, "Text-To-Speech Data Augmentation for Low Resource Speech Recognition," arXiv:2204.00291 [cs.CL], https://arxiv.org/abs/2204.00291
- Anton Ragni, Kate M. Knill, Shakti P. Rath and Mark J. F. Gales, "Data augmentation for low resource languages," in Proc. Interspeech, 2014, https://eprints.whiterose.ac.uk/152844/8/Ragni%20et%20al%202014%20Data%20augmentation%20for%20low%20resource%20languages.pdf
- M. J. F. Gales, A. Ragni, H. AlDamarki and C. Gautier, "Support Vector Machines for Noise Robust ASR," in Proc. ASRU, 2009, http://mi.eng.cam.ac.uk/~mjfg/gales_ASRU09.pdf
- Shaofei Xue, Jian Tang, Yazhu Liu, "Improving Speech Recognition with Augmented Synthesized Data and Conditional Model Training," Proc. International Symposium on Chinese Spoken Language Processing (ISCSLP), 2022, https://www.colips.org/conferences/iscslp2022/Proceedings/papers/ISCSLP2022_P008.pdf

Second, the reference to the classic paper by Larry Rabiner on HMMs should be

Lawrence R. Rabiner, A tutorial on hidden Markov models and selected applications in speech recognition. Proceedings of the IEEE, 77(2):257–286, 1989.

This will eliminate the cryptic "(R. 1989)" reference.


**Audience:**

There are elements of the paper that would be of interest to parts of TMLR's audience. Specifically, the concatenative synthesis method described in this paper does appear to be a simple and effective method for improving the performance of low-resource speech recognition models.


**Claims And Evidence:**

While the experimental results appear to be accurate, the submission is not sufficiently clear and convincing to appear in TMLR.

Regarding clarity, even following revision, the submission has a number of problems.

1. Non-standard terminology. Like reviewers cbZY and yJR9, I find the term "meta-audio set" to be opaque. It would be much clearer for the authors to state that they are performing concatenative text-to-speech and selecting the units for concatenation based on the characteristics of the language and the amount of available data. In Sections 3.4 and 3.5, the submission should consistently use the standard term "segmentation" rather than "time slice" or "partition."

2. The theoretical discussion obscures what is actually being done. A substantial amount of space in the paper is devoted to the Bayesian sampling perspective, but as reviewer cbZY notes, much of what is laid out in the paper is standard material in speech recognition and there do appear to be some errors in the derivations, which means that the submission is also not as convincing as it should be. The third line of Eq. 2 is only correct if the mapping from $y$ to $a$ is deterministic, but this is not the case as in many languages there are homographs. For example, the word "lead" in "Please get the lead." would have one pronunciation if it were uttered by a dog trainer (L IY D) and another if it were uttered by a chemist (L EH D). Furthermore, Section 3.4 implies that the proposed algorithm is attempting to maximize $P(x|a)$, but this is not actually the case because when there are multiple instances of a given unit, $a$, in the database used for concatenative synthesis, the realization of $a$ is selected at random. If one were actually attempting to maximize $P(x|a)$, more care would be taken in the selection. For example, where available realizations from a single speaker could be taken, or at least realizations from speakers having similar pitch ranges. The literature on concatenative TTS is full of various metrics aimed at making the generated audio sound as realistic as possible, which is closely related to maximizing $P(x|a)$. Much of the appeal of the proposed algorithm is that such care may not be needed, as it might not make much difference to the performance of the ASR system being trained, although Table 6 does show that energy normalization matters in some cases.